# Self-Reported Assessment of Empathy and Its Variations in a Sample of Greek Social Workers

**DOI:** 10.3390/healthcare9020219

**Published:** 2021-02-17

**Authors:** Maria Moudatsou, Areti Stavropoulou, Athanasios Alegakis, Anastas Philalithis, Sofia Koukouli

**Affiliations:** 1Department of Social Work, Hellenic Mediterranean University, 714 10 Heraklion, Greece; moudatsoum@yahoo.gr; 2Department of Nursing, University of West Attica, 122 43 Athens, Greece; aretistavropoulou@gmail.com; 3Department of Toxicology, University of Crete, 700 13 Heraklion, Greece; alegkaka@uoc.gr; 4Department of Social Medicine, University of Crete, 700 13 Heraklion, Greece; philal@uoc.gr

**Keywords:** empathy, healthcare professionals, social workers, empathy scale for social workers (ESSW), Greece

## Abstract

The aim of the study was twofold: (a) to examine the way in which professional social workers perceive and apply in their practice the concept of empathy; (b) to explore sociodemographic factors, education/special training and work characteristics associated with their empathic skills. This is a cross-sectional study with a purposive sample of 203 Greek social workers. For the assessment of empathy, the Empathy Scale for Social Workers (ESSW) was used. The sample consisted mainly of female social workers with a mean age of 43.8 years. More than 70% of them were practicing the profession for more than 10 years. Nearly one-third participated in psychotherapy courses, and only half of them have been certified. On average, they reported high levels of empathy. Initial univariate analyses showed that empathy scores were significantly higher for older social workers, married, the more experienced, those who referred to working experience with disabled people or people having problems with substance use and the professionals who had obtained a certification in psychotherapy. ‘Having a middle work experience of 10–19 years’ was a significant correlate in all scales and related negatively to empathy indicating a burnout effect. The implications for social work education and future training are discussed.

## 1. Introduction

Empathy is a fundamental concept within the psychodynamic, behavioral, and person-centred approaches because of its core role in the development of a therapeutic relationship with the clients [1,2]. It is the ability of perceiving and understanding the emotional state and ideas of another person by imagining what it might feel like to be them and living their life temporarily [3,4]. Rogers [5,6] explains empathy as sensing the client’s private world as if it were your own, but without ever losing the ‘as if’ quality. Although initially it was approached exclusively as a cognitive concept, gradually the affective and the behavioral dimensions were added [2]. The ability to understand a certain feeling of a person and conveying this situation to that person is called empathy. Empathy is the act of being aware of and responding to another person’s needs, feelings and situation [7]. In spite of its significance for social workers it is an under-investigated area in the field of social work practice. 

### 1.1. The Significance of Empathy in Social Work Practice

The concept of empathy is a core value for healthcare professionals in general and social workers in particular [4,8,9,10]. According to the Statement of Ethical Principles, published by the International Federation of Social workers, all social workers have the responsibility to respond with compassion, care, and empathy when working with people [11]. Social work professionals must have empathic skills in order to approach and sensitize the recipients of their services, as they are dealing with people in vulnerable situations in the daily practice of their work [4,12,13]. Empathy connects the social worker with people from a large variety of groups and different socio-cultural backgrounds. Being able to understand and sharing the feelings of these individuals is a difficult task, since empathy requires cognitive, emotional, behavioral, and moral capacities to understand and respond to the suffering of others [14,15]. It is however an extremely necessary task and an indicator of professional competence, as throughout this communication vulnerable groups are enabled to function more effectively in fulfilling their role [11,16].

Empathy is central to build a therapeutic relationship [17] and an important tool for a positive therapeutic intervention [9,18]. This relationship is the “medium which facilitates exploration of issues, provides hope and nurturance and is the channel through which strategies for intervention are introduced” [4]. Thanks to the fact that empathy possesses therapeutic potential, a social worker is able to provide a calm and appreciative approach to his/her applicant, and this weakens the break-off phenomenon of the applicant, thus strengthening the process of problem solving [19]. A social worker who pays empathic attention to social and economic inequalities is more eligible to influence effective social and economic justice and general wellbeing [20,21]. Empathy has a therapeutic effect because every empathic response conveys to the clients that the social worker not only supports them but he/she tries to understand his/her perception of distress [22]. The empathic approach strengthens the therapeutic relationship, breaks down defensive behavior and enables clients’ engagement in the process [4].

Additionally, empathy is considered to have a key role in a holistic assessment and effective intervention because it might be an important tool for collecting accurate information from the client [18]. For example, in child and family social work ‘Skills in forming relationships are fundamental to obtaining the information that helps social workers understand what problems a family has and to engaging the child and family and working with them to promote change’ [23]. Social workers who meet the recipients of services with an empathic approach have more possibilities in managing social change, because the clients feel more secure in trusting them, sharing their problems and helping them become aware of their needs [2,24].

### 1.2. Conceptualization of Empathy in Social Work and Assessment Tools 

During the last decade, the efforts of social workers to find instruments to quantify and operationalize empathy have been based on a holistic and comprehensive conceptualization of the term including not only the affective and cognitive components, but also the behavioral one [25]. An empathic social worker experiences an effect, processes it, and then takes action. Therefore a ‘social work model’ of empathy encompasses the notions of ‘social empathy’ (the ability to understand the social context of other people’s living conditions) and ‘empathic action’ (voluntarily taking action in response to cognitive processing and affective reaction) [20,21,25]. 

Various self-reported measures were developed in recent years for social workers [25], the “Empathy Scale for Social Workers” (ESSW) being one of them. It was developed by King and Holosko [24], and is a tool for assessing empathy in social work practice, for both professionals and students. It is based on the Jefferson Scale of Physician Empathy (JSE) [26] created by Hojat et al. [27], which is used to measure empathy in various groups of health care professionals and students [28]. The ESSW is based on the assumption that empathy is a complex concept with cognitive, affective and behavioral components [24]. The cognitive dimension involves interpersonal sensitivity and perspective taking. The affective component refers to caring and congruence. Finally, the behavioral dimension encompasses altruism and the development of a therapeutic relationship [24].

### 1.3. Empathy Variations

Nevertheless, empathy is emphasized as something important in social work profession, and it is not prioritized within the daily routine of a social worker. Practitioners’ ability to empathize with others is related to internal and external factors. Pressure of time, resource constraints, workplace frustrations, bureaucratic ethos, cultural, racial, ethnic and socio economic differences are external factors that put limits on empathy [4]. For example, many studies indicated that the more experienced a social worker is, the more likely he/she is to be empathic and to have less psychological distress [15,29,30]. Additionally, female social workers have more empathic skills than men [31]. Among the internal factors that prevent a social worker from adopting an empathic process are the belief in the professionals’ superiority and the fear of losing his/her power [4]. In addition, social workers’ personal problems reduce their empathic skills. Social workers ought to take care of themselves personally and professionally, especially in their workplace, in order to show empathy to their clients [11].

The personal development and self-awareness of a social worker are fundamental in dealing with their clients [32,33]. Social workers who are aware of their feelings and needs are more eligible to show empathy to their clients [34]. In a Swedish study among social workers, most of the participants suggested that whenever they were aware of their feelings and reactions, they did not hesitate to ask for help and to talk about their professional difficulties [29]. Regular supervision is also necessary for all social workers in order to be able to process their own feelings and to deal with empathy [29].

Social workers are less empathic when they feel stressed because of heavy work, time pressure or when they do not have the knowledge to deal with a difficult situation. Work related stress puts barriers on empathic attention because stress produces a dysfunction and reduces control in the work process [29]. In addition, the huge number of clients in daily social work practice and the lack of training programs are reasons that impede empathy [2,35].

### 1.4. Aim of the Study

Exploring what social workers say about empathy and the factors influencing it is very important for our understanding of the concept. Although empathy is considered a critical and essential ability for effective social work practice, empirical evidence on empathy remains scarce [9,10,15,36]. Past research has examined it in helping professionals, but few studies have focused on social workers [31], and these were mainly based on samples of undergraduate students and rarely professionals. Therefore, the objective of this study is to explore the levels and dimensions of empathy as well as to investigate the sociodemographic factors and work characteristics associated with it in social work professionals.

## 2. Materials and Methods

### 2.1. Design

This is a descriptive study with a cross-sectional design carried out in a purposive sample of Greek social workers.

### 2.2. Sample and Procedure

All social workers of the Region of Crete, employed in social services of the public and private sectors, constituted the study population. The selection criterion was having at least two years of working experience as a social worker after graduation.

The study sample was recruited both through the National and Local Associations of Social Workers and directly from the services we contacted and which employ social workers in the Region of Crete. During data collection there was a continuous check to avoid overlaps. After finding the total number of social workers employed in those services, the heads of the services were asked for their permission in order to get in touch with the social workers of their departments. Although more time-consuming, it was deemed appropriate to have more personal contact with every potential participant in order to increase the participation rate. Therefore, each potential participant was contacted personally to give his/her initial consent and then received formal invitations to participate in the study by a letter or email, in which the purposes of the research were described in detail.

The data were collected by the first author, a professional social worker herself, systematically using a questionnaire. A pilot study was carried out in a small sample of 10 social workers who were employed in social services outside of Crete. The pilot survey preceded the main survey in order to determine the feasibility of the study and the suitability of the questionnaire used. These social workers were excluded from the final sample.

From the 259 professionals initially contacted, 203 participated in the survey (response rate 78.4%). The vast majority of them were employed in various social services. Only three respondents (1.47% of the total) were unemployed at the time of the survey. The questionnaires were administered to the participants either personally or via conventional mail.

#### 2.2.1. Ethical Considerations 

The study’s protocol was approved by the Research Ethics Committee of the Hellenic Mediterranean University (Ref. No. 781/6.12.2017). Approval was also gained from the Regional Authorities of Crete before data collection. Relevant information about the study and anonymity and confidentiality issues were explained to the participants before data collection. The voluntary nature of the research and their right to withdraw from the study at any time without any drawback were also stressed.

#### 2.2.2. Measures

The survey questionnaire consisted of four parts: demographics, job characteristics, special training and assessment of empathy.

#### 2.2.3. Demographics

In the first part a Demographic Information Questionnaire was used consisting of five questions assessing participant’s age, gender, marital status, number of children, and educational level.

#### 2.2.4. Job Characteristics

In the second part, respondents, both employed or unemployed during the survey period, were asked to provide the total number of years of social work practice after graduation and additional information regarding their actual job placement, such as their area of practice (public or private sector) and the social groups they had worked with throughout their entire professional career until the day of the survey.

### 2.3. Special Training

In the third part, questions on their participation in counseling or psychotherapy courses and holding or not a certification in counseling/psychotherapy were included as well. They were also asked if they participated in a personal development or self-awareness program, personal and social skills training program, group therapy, psychotherapy or in supervision programs designed for professional social workers. 

### 2.4. Assessment of Empathy

The empathy of professionals was evaluated with the “Empathy Scale for Social Workers” (ESSW). The ESSW is a 41-item self-report inventory designed to assess empathy in social work practitioners [24]. Permission was granted by the authors and the Greek version of the scale was developed using the method of front and back translation. The first and the last author translated the questionnaire from English to Greek and one bilingual psychologist did the backward translation. Differences in translation were discussed and resolved through consensus. The items describe thoughts, feelings, and actions involved in the use of empathy in social work practice. Participants responded to each one of them on a Likert-type scale with answers ranging from 1 to 5 (never, rarely, sometimes, often, and always). The ESWW contains four reverse scored items. Higher scores represent higher levels of empathy.

According to the authors there are two approaches that could be used in analyzing the data: a) first, a total score can be calculated by adding the scores of all the 41 items, with higher scores representing higher levels of empathy (Total_41). b) In addition, three separate scores, one for each subscale, can be derived from a factor analysis. These subscales reflected three different dimensions of empathy: the first factor with 10 items named “a compassionate contextual assessment” (CCA), describes a framework for understanding the experience of receiving and delivering social work services (e.g., ‘I try to take a client’s cultural context into account when working with them’). The second factor with 8 items named “an accepting and attentive collaborative inquiry” (ACI), describes the relationship style and quality inherent in direct social work practice (e.g., ‘It is important for my clients to know that I care about them’). The final factor consisting of 4 items was named ‘‘intrinsic helping and emotional support’’ (IHS), and reflected behavioral expressions of caring and altruism in an empathic helping experience (e.g., ‘Helping clients is rewarding in and of itself’). In addition, a total score can be derived from these three separate scores (Total_22).

### 2.5. Statistical Analysis

Counts and percentages (%) were used to describe discrete, nominal and ordinal data, while means and SDs were used to describe continuous data. Pearson’s chi-square was applied for examining bivariate association and differences in proportions of discrete variables. One way ANOVA and independent samples t-test was applied for measuring mean differences between >2 and 2 groups, respectively. Reliability of the scales was estimated using Cronbach’s alpha statistics. Multiple linear regression models were also applied, with backward selection, using empathy scales as dependent variables (CCA, ACI, HIS, Total_22 score and Total_41 score) and as an independent set of variables with a p-value less than 0.200 as resulted from univariate analyses. IBM SPSS version 24.0 was used for statistical analysis.

## 3. Results

### 3.1. Descriptive Statistics

#### 3.1.1. Sociodemographics of the Sample and Work/Special Training Characteristics

The study sample consisted mainly of female social workers (92.6%). The mean age of the participants was 43.8 ± 9.7 and most of them were married, parents and with a bachelor degree from a Technological Educational Institute or University. A smaller percentage had a Master’s Degree or PhD (24.9%). In addition, the majority of the sample (>70%) had a lengthy work experience of more than 10 years as social workers, mainly in the public sector. Nearly one-third of the professionals participated in counseling/psychotherapy courses, and only half of them have been certified. More than half of the sample had participated in personal development seminars, or attended a supervision program (Table 1). In Figure 1 the type of users to whom these professionals provided social services are presented in descending order. Most of the served persons belonged in the Elderly group, followed by the mentally ill and the disabled.

#### 3.1.2. Assessment of Empathy

Descriptive statistics of the two ESSW scales and the subscales are presented in Table 2. On average, the sample population reported fairly high levels of empathy. For the 41-item scale the highest possible score is 205 and the scores of the respondents ranged from 140 to 193 (M = 170, SD = 11.0). For the 22-item scale the highest possible score is 110 and the total scores ranged from 70 to 110 (M = 93.9, SD = 7.3). The internal consistencies, assessed with Cronbach’s alpha, were 0.804 for the 41-item scale, 0.802 for the 22-item scale and lower for the three subscales (CCA = 0.675, ACI = 0.673 and IHS = 0.649).

### 3.2. Bivariate Associations of Empathy with Sociodemographic Variables, Work Characteristics and Special Training

Table 3 shows the association between empathy and sociodemographic variables. Gender and education levels did not have any effect on empathy scores. Regarding age groups, significant differences were observed between CCA, ACI and the 22-item scale. The older participants (over 50 years of age) had higher empathy scores, while the middle-age groups (30–39 and 40–49) had lower values. In addition, the ACI and the IHS mean values were found to differ significantly between family status groups. The participants who were married scored higher on ACI compared to the unmarried and the divorced/widowed of the sample, while the singles had higher mean values on IHS compared to the other two groups. Additionally, parenthood did not show any significant effect on the empathy scales resulting in p-values greater than 0.400.

Table 4 presents the differences in empathy scores between the types of vulnerable groups served by the participants. Significantly higher scores in empathy were found in CCA between respondents who worked with individuals struggling with substance abuse compared to those who did not. Differences were also observed on CCA and the two total score scales between those professionals who worked with disabled vs. those who did not have this experience.

In Table 5 the bivariate associations between empathy and various job characteristics are shown. More years of work experience (20+) were significantly related with higher scores on empathy scales (ACI, Total_22 and Total_41). In addition, those with less than 10 years of experience scored higher on the IHS scale (borderline p value). Other job characteristics, such as the sector of their work (public/or private), or the number of recipients of their services did not seem to have an impact on empathy scores (results not shown here).

As regards special training variables, attending personal development courses or participating in supervision programs was not associated with levels of empathy. However, being certified as a psychotherapist made a difference, as those who gained a certificate scored higher than those who replied negatively on almost all scales of measurement.

To identify the antecedents of empathy, we used multiple linear regression analysis with a backward selection. Empathy scores (three subscales scores and two overall scores) were the dependent variables, whereas a different set of independent variables was introduced for each model (Table 6).

For the CCA, the subscale score was dependent, and age, the vulnerable groups participants had worked with, length of work experience, and being a certified psychotherapist were introduced as independent variables. Three significant correlates of empathy emerged: a) “middle” work experience of ‘10–19’ years with a negative effect on CCA (b = −1.85, p = 0,012), b) working with children/teenagers and c) having work experience with persons with a disability, both b and c with a positive effect on CCA (b = 1.69, *p* = 0.04 and b = 1.77, *p* = 0.05, respectively). Furthermore, ‘being a certified psychotherapist’ was marginally significant. For the ACI subscale the initial set of independent variables included age, length of work experience, marital status, two vulnerable groups (elderly and disabled) and being certified as a psychotherapist. The final model comprised two significant antecedents of empathy: those with ‘middle’ work experience who apparently were less empathic than social workers with fewer years of experience (b = −1.37, *p* = 0.025) and those professionals working with the elderly who showed higher levels of empathy compared to those who did not have work experience with this vulnerable group (b = 1.76, *p* = 0.022). The initial set for the IHS subscale score as dependent included age, marital status, and working with children/teenagers, or immigrants. Only two groups of work experience ‘10-19 years’ and ‘20+ years’ had a significant effect on this subscale, both being negatively associated with empathy (b = −0.88, *p* = 0.059 & b = −0.93, *p* = 0.039, respectively), although for the first group the p-value was marginally significant. It was rather expected that those with less than 10 years as professionals reported more behavioral expressions of caring and altruism in an empathic helping experience compared to the other two groups.

For the total scales (total_22 and total_41) the initial set of variables was almost the same (including age, having experience with persons with disabilities, years of working experience, and being certified as a psychotherapist), except for ‘attendance to personal development seminars’, which was also added in the group of independent variables for total_41. In both models resulting from the analyses, the group of professionals with the middle number of years of work-experience (‘10–19 years’) was the only significant correlate, having a negative effect (b = −3.28, *p* = 0.014 and b = −4.28, *p* = 0.04) on empathy scores. Variance inflation factor (VIF) statistics did not indicate that multicollinearity was a significant problem.

## 4. Discussion

This cross-sectional study was carried out with the aim of examining the levels and types of empathy in a sample of Greek social workers employed in various health and social care services, as well as analyzing wider factors that might influence it, such as the participants’ sociodemographics, work, and extra training characteristics. It is innovative in exploring empathy in professional social workers, a previously understudied research area, since, to our knowledge, past research focused for the most part on samples of undergraduates and qualitative data.

### 4.1. Empathy Levels

In general, our data showed, as indicated by the overall and the subscales mean scores, that the majority of social workers reported high levels of empathy, suggesting a high degree of professional competence in these practitioners. In other studies, low levels of empathy were identified [37,38]. Findings of various studies on empathy assessment are not directly comparable as they may be qualitative or quantitative, use dissimilar samples or different measurement tools, or be conducted in different settings. In addition, the social desirability bias, common in self-reports, might explain the high levels of empathy [39]. The internal consistency of the three subscales is considered ‘satisfactory’ (close to 0.7) and of the overall scales ‘very good’ (>0.80) and comparable to those of other studies [1,17] using the same measure of empathy (ESSW).

### 4.2. Gender, Age, Family Status and Empathy 

According to the regression analysis none of the sociodemographic characteristics emerged as a significant correlate of empathy. Some of the results of the univariate analyses are discussed below.

In this sample, no statistically significant differences were found between genders. It is possible that this finding is due to the under-representation of male social workers and the fact that social work is a female-dominated profession. Previous studies in helping professionals are not consistent regarding the effect of gender. Some researchers found that men were less empathic than women [13,17,40] and others are in line with our finding that gender is not associated with empathy [31].

On the contrary, age was a significant correlate of empathy. Participants aged 50 years and older reported significantly higher levels of empathy than those under 50. This finding confirms prior research conducted in social workers’ samples [15,17] and other health professionals [41,42,43]. It is interesting that Oh et al. [44] in their recent study examining changes in empathy in six longitudinal samples deriving from the general population, found that empathy increased across the lifespan, particularly after age 40. It is noteworthy that for CCA and the total-22 scale, empathy scores are higher for those ‘under 30’, slightly decline in the second age group (30-39), and increase again after 40 years and above.

With respect to family status, married respondents reported higher empathy scores on ACI compared to the other two groups. Similar results were found in samples of other health professionals [45]. It is very likely that coexistence with other family members (e.g., partners, young children, teenagers, elderly people) helps in learning to listen to other people’s needs or in managing complex situations that occur in the family environment and, thus, sharpen their empathic skills. For IHS, results were different as the unmarried professionals scored higher on empathy implying that helping others may be more important for them in terms of moral and emotional rewards, while married people usually acquire these rewards through their family and children. 

#### 4.2.1. Job Characteristics and Empathy

The results of univariate analyses showed that empathy scores (ACI and the two total scales) increased with the number of years of work experience. These findings are also confirmed in other helping professions [46,47] and are undoubtedly linked with the age results described above, as older people are usually more experienced professionals. Age and years of work experience are also related to other variables indicating a better quality of working life and suggesting better empathic skills. For example, older social workers and other helping professionals reported more compassion satisfaction, i.e., the positive feelings one obtains from being involved in the healing process of other individuals [48,49], lower stress in the workplace [48], and lower rates of burnout and compassion fatigue [50,51]. However, among the three groups referring to the length of work experience, only the ‘middle’ one (10 to 19 years) emerged as a significant antecedent in all scales and was associated negatively with empathy. This finding may indicate a “burn out” factor in this group, as it consists for the most part of social workers in their ‘30s and ‘40s with a heavy workload in addition to family responsibilities. In Lazo’s qualitative study [29] it was mentioned by the respondents that occupational stress could negatively affect their empathic attitude.

Regarding professional experience with vulnerable groups, the regression analysis generated three of them as associated significantly and positively with empathic skills: disabled, children/adolescents (CCA subscale), and elderly people (ACI subscale). As people with disabilities are a vulnerable group with many difficulties, a professional cannot work with them without having empathic skills. For example, for people with intellectual disabilities, who constitute the largest part of this wider group of disabled users of our sample, professionals are often their greatest source of emotional support as their networks are very restricted. In addition, some of them are less able to communicate feelings clearly, or may do so through challenging behaviors. Therefore, they rely on empathic carers to interpret their needs and respond accordingly [52]. On the other hand, the professionals of the sample may have reinforced their empathic skills through the provision of services to people with disabilities, as it is usually very difficult to intervene in these groups, or to persuade them to do things. Unlike working with any other group, these social workers have to come to terms with more irreversible situations.

Moreover, the work with families, children and adolescents at risk, may involve dealing with people in complex situations (e.g., victims of abuse, domestic violence, extreme poverty). The social worker must have the ability to focus on difficult issues, while combining this with the ability to empathize with the parent. Finally, the elderly are also a group with particularities as their declining health and functioning may also impact their behavior, mood, and personality and impede their willingness to participate in social work intervention programs. Providing social services to older adults and building a successful therapeutic relationship with them often requires unique communication skills and strategies.

#### 4.2.2. Psychotherapy Training, Supervision, and Empathy

Postgraduate education alone did not make a difference with regards to empathy. However, those who not only attended but also completed a more specialized psychotherapy program by obtaining the corresponding certification reported higher scores on CCA and the total scale. Having a certificate in psychotherapy was also one of the antecedents of CCA. This result emphasizes the positive effect of psychotherapy programs on empathic skills [53]. It is well known that this kind of certification requires personal psychotherapy counseling and training in personal and social skills, attending a well-defined curriculum with a given duration, and specific criteria that should be fulfilled. In addition, a certified special training includes many experiential exercises that in themselves enhance empathy.

Finally, a high percentage of respondents said that they did not have supervision, implying its lack in Greece. Personal psychotherapy and professional supervision are keys to enhancing empathy. Some social workers attended personal development courses through individual or group psychotherapy, but many of them did not complete them. A certificate from a psychotherapist is required for the completion of a supervision program.

#### 4.2.3. Study Limitations

Results should be interpreted while taking into account study strengths and limitations. One strength is the assessment of empathy in a group of social workers, using an adequate sample size that covers almost all professionals of a concrete geographical area in Greece. However, a larger, randomly selected national sample would have been more representative and would increase generalizability. Additional limitations may include the low representation of male respondents, and the weaknesses of self-reports. Current findings also suggest that in assessing empathy it is more appropriate to use a mixed-methodology of qualitative and quantitative measures than using only self-report measures. This way, social desirability bias, which prevents people from giving truthful answers to survey questions, leading to skewed results, would be minimized. According to Sassenrath [39], self-reported empathic responses are confounded by social desirability. Being empathic and showing empathic behavior in response to others’ needs and conditions are socially desirable. Her findings have documented that empathic responses are, to a considerable degree, confounded with socially desirable responses.

## 5. Conclusions

Considering the results of the present study, a gamut of factors can exert a positive influence on increased empathy among professional social workers such as being married, older, having more years of work experience, or working with specific vulnerable groups. In addition, current findings point out the significance of training in psychotherapy and the importance of supervision programs for the enhancement of professionals’ empathy skills. Self-other awareness, perspective taking, and emotion regulation do not come automatically, but can be learned through education. Empathy can be taught, increased, and refined, so that social workers might become more skillful and resilient [25,54]. Cultivating empathic attitudes through these processes protects professionals from compassion, fatigue, and burnout and helps them develop sufficient levels of compassion satisfaction [55,56].

It is also imperative that social workers be better prepared through their undergraduate education and training. In Greece, as in other cultural contexts [17,29], the subject of empathy is not covered in depth within social work studies. Special courses should be part of the curriculum in order to enable students to develop their emotion management, social skills, and wellbeing and enhance their professional empathy. Innovative teaching methodologies should be used to provide social workers with empathetic techniques enabling them to incorporate these skills into their professional attitude [15,57,58].

It continues to be a challenge for educators to measure empathy and predict whether certain individuals would need additional training in order to develop better empathy skills. Using various ways of assessing empathy, not only through self-reports, but also through mixed methods and in various settings, will help us better understand how empathy is incorporated into social work practice, with important implications for the way we develop social work education and deliver services.

## Figures and Tables

**Figure 1 healthcare-09-00219-f001:**
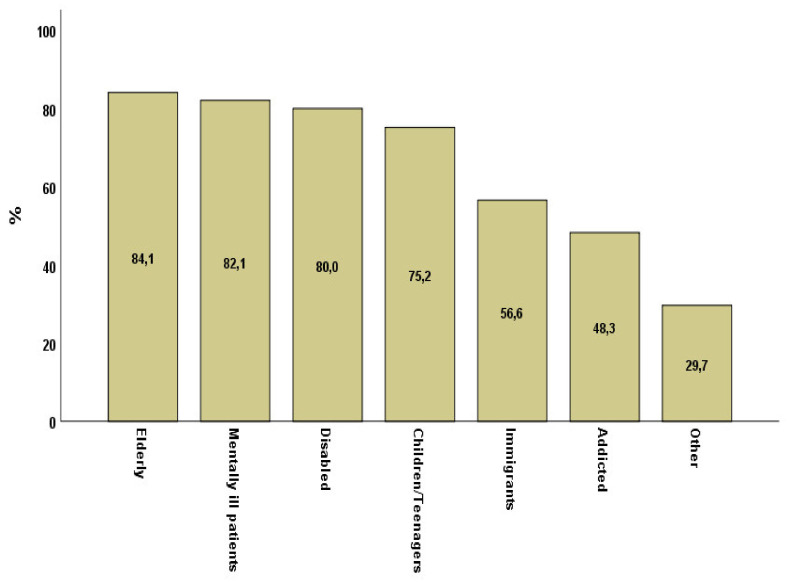
Categories of users of services.

**Table 1 healthcare-09-00219-t001:** Sociodemographic, work, and education/training characteristics of the sample.

		*N*	%
**Gender**	Female	187	92.6
	Male	15	7.4
**Age groups**	< 30	12	6.5
	30–39	54	29.2
	40–49	57	30.8
	50–59	56	30.3
	60+	6	3.2
**Marital status**	Single	49	24.4
	Married	131	65.2
	Divorced/Widow	21	10.5
**No of children**	0	53	27.2
	1+	142	72.8
**Educational level**	Technological Education	135	67.2
	University	11	5.5
	MA	50	24.9
	PhD	5	2.5
**Work experience as a SW**	<10	54	28.1
	10-19	69	35.9
	20+	69	35.9
**Job sector**	Public	109	55.9
	Private	49	25.1
	European programs	23	11.8
	Other	14	7.2
**Counseling/psychotherapy**	No	132	66.0
**courses**	Yes	68	34.0
**Certification in**	No	163	84.9
**counseling/psychotherapy**	Yes	29	15.1
**Personal development**	No	84	42.2
**/self-awareness programs**	Yes	115	57.8
**Supervision program**	No	131	65.2
	Yes	70	34.8

**Table 2 healthcare-09-00219-t002:** Descriptive statistics for the ESWW scale (total scores and its components).

Scales	Title	Mean	SD	Median	Min	Max	Cronbach’s Alpha
CCA	Compassionate contextual assessment	42.8	4.0	43.0	31	50	0.675
ACI	Accepting and attentive collaborative inquiry	33.3	3.4	33.0	23	40	0.673
IHS	Intrinsic helping and emotional support	17.9	2.0	18.0	12	20	0.649
Total_22	CCA, ACI, IHS (22 items)	93.9	7.3	94.0	70	110	0.802
Total_41	All items (41)	170.0	11.0	171.0	140	193	0.804

**Table 3 healthcare-09-00219-t003:** Association of empathy with sociodemographic variables—means (SDs).

	CCA	ACI	IHS	Total_22	Total_41
Gender					
Men	43.1 (3.7)	33.1 (3.1)	18.3 (1.4)	94.6 (6.1)	173.1 (7.9)
Women	42.7 (4.0)	33.3 (3.4)	17.8 (2.0)	93.8 (7.4)	169.7 (11.2)
*p*	0.715	0.898	0.333	0.688	0.254
Age groups					
<30	43.3 (3.4)	31.3 (3.2)	18.3 (1.6)	92.9 (5.5)	167.9 (8.4)
30–39	41.9 (3.7)	32.9 (3.3)	17.9 (1.9)	92.7 (6.7)	169.1 (10.3)
40–49	42.1 (4.0)	33.0 (3.5)	17.7 (1.9)	92.8 (7.6)	168.6 (11.4)
50–59	43.9 (4.1)	34.1 (3.2)	17.7 (2.3)	95.8 (7.9)	172.8 (11.6)
60+	44.5 (4.3)	36.3 (2.1)	18.2 (2.1)	99.0 (5.5)	174.0 (7.2)
*p*	0.035	0.007	0.899	0.047	0.189
Family status					
Single	43.1 (3.5)	32.3 (3.5)	18.1 (1.5)	93.6 (6.5)	168.8 (10.6)
Married	42.7 (4.1)	33.7 (3.2)	17.9 (2.0)	94.3 (7.3)	170.8 (11.0)
Divorced/Widow	42.1 (4.4)	32.3 (4.0)	16.9 (2.5)	91.3 (9.1)	167.3 (12.5)
*p*	0.662	0.029	0.041	0.208	0.299
Education					
University	42.7 (3.9)	33.1 (3.5)	18.0 (2.1)	93.8 (7.4)	169.8 (10.9)
MA/PhD	42.9 (4.2)	33.5 (2.9)	17.7 (1.7)	94.1 (7.1)	170.5 (11.3)
*p*	0.675	0.464	0.376	0.748	0.668

**Table 4 healthcare-09-00219-t004:** Association between working experience with various groups of users and social workers’ empathy—mean scores (SDs).

	CCA	ACI	IHS	Total_22	Total_41
Elderly					
Yes	42.9 (4.0)	33.4 (3.2)	18.0 (2.0)	94.4 (6.9)	170.1 (11.2)
No	42.4 (4.5)	32.4 (3.6)	17.6 (2.2)	92.4 (8.7)	169.8 (11.5)
*p*	0.556	0.184	0.395	0.244	0.908
Childhood/Adolescence					
Yes	43.1 (4.0)	33.3 (3.3)	17.8 (2.1)	94.3 (7.4)	170.2 (11.7)
No	42.0 (4.1)	33.1 (3.3)	18.4 (1.5)	93.4 (6.9)	169.6 (9.6)
*p*	0.147	0.654	0.147	0.544	0.762
Mental illness					
Yes	42.9 (4.1)	33.3 (3.3)	17.9 (2.0)	94.1 (7.1)	170.4 (10.7)
No	42.6 (4.0)	33.3 (3.4)	18.0 (1.9)	93.9 (7.9)	168.7 (13.6)
*p*	0.707	0.947	0.780	0.919	0.501
Disability					
Yes	43.1 (4.2)	33.5 (3.1)	18.0 (1.9)	94.6 (7.0)	171.2 (10.2)
No	41.7 (3.4)	32.4 (3.7)	17.6 (2.3)	91.7 (7.8)	165.8 (13.8)
*p*	0.041	0.103	0.295	0.049	0.020
Addiction					
Yes	43.6 (4.0)	33.4 (3.5)	17.7 (2.2)	94.8 (7.7)	171.2 (11.3)
No	42.1 (4.0)	33.1 (3.1)	18.1 (1.9)	93.4 (6.8)	169.0 (11.1)
*p*	0.024	0.608	0.261	0.242	0.227
Immigration					
Yes	43.2 (4.0)	33.0 (3.1)	17.7 (2.1)	93.9 (7.3)	169.1 (11.6)
No	42.4 (4.1)	33.6 (3.5)	18.2 (1.9)	94.2 (7.3)	171.3 (10.6)
*p*	0.256	0.240	0.187	0.791	0.238

**Table 5 healthcare-09-00219-t005:** Association between work experience and special training with empathy scores.

	CCA	ACI	IHS	Total_22	Total_41
Work experience					
<10 years	42.5 (3.6)	32.5 (3.6)	18.4 (1.6)	93.5 (7.0)	171.6 (9.8)
10–19 years	42.2 (3.7)	32.8 (3.2)	17.5 (1.9)	92.6 (6.7)	169.8 (10.1)
20+ years	43.6 (4.1)	34.2 (2.8)	17.7 (2.2)	95.6 (7.4)	174.7 (9.4)
*p*	0.066	0.009	0.056	0.039	0.039
Certification in psychotherapy					
Yes	44.8 (3.6)	34.4 (3.0)	18.2 (2.2)	97.4 (7.3)	174.2 (10.6)
No	42.4 (4.0)	33.1 (3.4)	18.0 (1.9)	93.4 (7.2)	169.4 (11.1)
*p*	0.004	0.037	0.309	0.009	0.033
Personal development/self-awareness					
Yes	42.5 (3.9)	33.2 (3.3)	17.8 (2.0)	93.5 (7.1)	169.1 (11.0)
No	43.1 (4.1)	33.4 (3.5)	18.1 (1.8)	94.5 (7.7)	171.2 (11.1)
*p*	0.299	0.730	0.395	0.346	0.178
Supervision					
Yes	42.9 (4.1)	33.1 (3.3)	17.9 (2.1)	93.9 (7.5)	169.7 (11.4)
No	42.7 (3.9)	33.2 (3.4)	17.9 (2.0)	93.8 (7.3)	170.0 (10.9)
*p*	0.746	0.840	0.917	0.912	0.837

**Table 6 healthcare-09-00219-t006:** Correlates of empathy as derived from multiple linear regression analysis with backward selection.

Empathy Scales	Independent variables	b ^1^	β ^2^	p ^3^	95%LB ^4^	95%UB ^4^
CCA	Length of work experience (10–19 years)	−1.88	−0.22	0.036	−3.65	−0.13
	Children/Teenagers	1.66	0.17	0.046	0.03	3.29
	Persons with disability	2.29	0.22	0.012	0.51	4.08
	Certified in psychotherapy	1.88	0.17	0.053	−0.03	3.80
ACI	Length of work experience (10–19 years)	−1.37	−0.20	0.025	−2.56	−0.18
	Elderly	1.78	0.21	0.022	0.26	3.29
IHS	Length of work experience (10–19 years)	−0.88	−0.22	0.059	−1.72	0.03
	Work experience (20+ years)	−0.93	−0.22	0.039	−1.81	−0.05
Total_22	Length of work experience (10–19 years)	−3.28	−0.21	0.014	−5.88	−0.67
Total_41	Length of work experience (10–19 years)	−4.28	−0.18	0.040	−8.36	−0.19

^1^ unstandardized coefficients, ^2^ standardized coefficients, ^3^
*p* ≤ 0.05, ^4^ 95% confidence interval for b (Lower and Upper Bound).

## Data Availability

The data presented in this study are available on request from the corresponding author. The data are not publicly available due to privacy reasons.

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
