# Peer review of "Self-Reported Assessment of Empathy and Its Variations in a Sample of Greek Social Workers"

_healthcare, 2021, doi:10.3390/healthcare9020219_

Round 1

Reviewer 1 Report

Dear Authors,

I have read the manuscript carefully and I think it presents a potentially interesting theme, the empathy in social working.

Despite the theoretical background, that is clearly presented and adequately articulated, I am sorry to tell you that the study presented is not consistent with the expectations; in fact, it is merely descriptive and does not add anything significant to the knowledge of empathy. Moreover, the aims indicated in the abstract create expectations that are not satisfied by the content of the manuscript: in fact, the point a) "to examine the way in which professional social workers perceive and apply in their practice the concept of empathy" is completely disregarded in the empirical study. 

I have appreciated the effort in recruiting the participants, and in conducting even a pilot study, but despite these premises, the use of a socio-demographic survey, with a unique measurement scale, does not allow to expand the knowledge about this issue.

Moreover, the analyses conducted does not permit to establish causal relationships or, almost, verify eventual effects of the independent variables on the dependent ones.

Subsequently, the discussion and the conclusion are ordinary and obvious.

Furthermore, I wonder why the empathy's subscales have almost acceptable alphas.

I regret to give you such a negative feedback, but I hope it could be useful for future research designs; associating other measure scales (i.e. personality traits or other psychological or working dimensions as burnout) or using more complex methods in data analysis, you could try to use the theoretical work already done for this manuscript.

I wish you the best!

Reviewer 2 Report

I really appreciated this paper and how the topic was addressed. I have only some minor comments especially on possible other analysis and on methods. I suggest also to have a revision of the manuscript by a native English speaker.

Methods

Alpha values are missing in the validation of the instrument and also possible correlations with other instruments in the past literature.

Gender cannot be considered as a possible factor influencing empathy: the numerosity of males and females are really not homogenous

Data analysis

Could be parenthood a variable that make more empathetic the person and not only age or familiar situation?

From the ANOVAs and correlations results, it should be better to run a regression model to identify the variables most predictive for empathetic attitude.

Discussion

Another possibility for future research recommendation is to assess also the emotive intelligence and also to compare empathy in groups of other types of work where there have social contacts but not necessary with vulnerable people.

Reviewer 3 Report

Congratulations on your article.

It offers an interesting introduction and presentation on what is the true essence of Social Work and its link with empathy. This offers consistency to the research work done.

Despite using another empathy measurement instrument (IRI), I recommend consulting Cuartero-Castañer's thesis, 2018, where the study of empathy in a sample of social workers is addressed.

You can compare your results with other research with the same instrument, or others, in the past academic research.
Your sample is very feminized. You can explain this result to conclude that Social Work is a feminized profession. Take care to generalize your results in gender because the name of males isn't significant.
It could be necessary that you introduce alpha values that now don't include.
I recommend you do extra analysis results, orientated to explore more consciously why social workers are empathic, which personal or professional/work characteristics can predict this empathy?

Round 2

Reviewer 1 Report

I have appreciated your effort in improving your manuscript following my comments.

Before publishing the manuscript, please change the use of the term 'predictors' and any reference to the predictions using 'antecedents' or similar, both in the abstract and in the text; we cannot infere causal relationships using merely the regression analysis, but a structural model should be used. 

Author Response

We thank the reviewer for this remark. We have now replaced the words ‘predict’ or ‘predictor/s’ (related to the results of the regression analysis in the text and the abstract) with the words ‘antecedents’ or ‘correlates’. Replacements are highlighted with yellow colour.